# Biocompatibility and Cytotoxicity Study of Polydimethylsiloxane (PDMS) and Palm Oil Fuel Ash (POFA) Sustainable Super-Hydrophobic Coating for Biomedical Applications

**DOI:** 10.3390/polym12123034

**Published:** 2020-12-18

**Authors:** Srimala Sreekantan, Mohd Hassan, Satisvar Sundera Murthe, Azman Seeni

**Affiliations:** 1School of Materials and Mineral Resources Engineering, Engineering Campus, Universiti Sains Malaysia, Nibong Tebal 14300, Pulau Pinang, Malaysia; mohdhassan@student.usm.my (M.H.); satisvar@student.usm.my (S.S.M.); 2Malaysian Institute of Pharmaceuticals and Nutraceuticals (IPHARM), National Institute of Biotechnology Malaysia, Ministry of Science, Technology and Innovation, Bukit Gambir, Gelugor 11700, Pulau Pinang, Malaysia; azmanseeni@usm.my

**Keywords:** hydrophobic coatings, MTT assay apoptosis, PDMS: SS coatings

## Abstract

A sustainable super-hydrophobic coating composed of silica from palm oil fuel ash (POFA) and polydimethylsiloxane (PDMS) was synthesised using isopropanol as a solvent and coated on a glass substrate. FESEM and AFM analyses were conducted to study the surface morphology of the coating. The super-hydrophobicity of the material was validated through goniometry, which showed a water contact angle of 151°. Cytotoxicity studies were conducted by assessing the cell viability and cell morphology of mouse fibroblast cell line (L929) and hamster lung fibroblast cell line (V79) via tetrazolium salt 3-(4–dimethylthiazol-2-yl)-2,5-diphenyltetrazolium bromide (MTT) assay and microscopic methods, respectively. The clonogenic assay was performed on cell line V79 and the cell proliferation assay was performed on cell line L929. Both results validate that the toxicity of PDMS: SS coatings is dependent on the concentration of the super-hydrophobic coating. The results also indicate that concentrations above 12.5 mg/mL invariably leads to cell toxicity. These results conclusively support the possible utilisation of the synthesised super-hydrophobic coating for biomedical applications.

## 1. Introduction

The healthcare industry is a behemoth that accounts for approximately 8.8 percent of gross domestic product (GDP) on average among the Organisations for Economic Co-operation and Development (OECD) nations in 2020. It is estimated that healthcare accounts for 17.0% of the nation’s 2020 GDP in the United States alone. Moreover, the GDP share is poised to rise in the near future due to the substantial increase in the aging population [1]. Unsurprisingly, the convoluted healthcare system takes a heavy toll on the environment as indicated by a study that showed that the healthcare sector contributes up to 8% of the total US greenhouse gas emissions [2]. In light of this, the fabrication of sustainable healthcare materials to minimize the negative environmental impacts of the healthcare sector has been taken up by research communities worldwide with great determination. However, despite the efforts to incorporate sustainable materials in biomedical applications, there currently exists a large deficit of sustainable super-hydrophobic coating materials for use in applications such as drug delivery and biomedical implants. One prime candidate for sustainable super-hydrophobic coating is palm oil fuel ash (POFA). The high silica (SiO_2_) content in POFA makes it an excellent sustainable candidate to extract SiO_2_ to be used as nanoparticles in super-hydrophobic coatings [3,4,5,6]. SiO_2_ is highly capable of increasing material surface roughness to obtain water contact angles (WCA) greater than 130° [7,8]. Furthermore, the SiO_2_ nanoparticles can enhance mechanical strength, thermal resistance, and high transparency to the super-hydrophobic coatings [2].

Table 1 showcases waste materials such as rice husk ash (RHA), sugarcane bagasse ash (SBA), fly ash (FA), paper sludge ash (PSA), and palm oil fuel ash (POFA) that have been consistently explored by various researchers. Among the tabulated materials, sustainable agricultural waste such as RHA, SBA, and FA can be found in abundance in Southeast Asian countries. Around 42.54 million tons of POFA, 13.81 million tons of RHA, and 9.75 million tons of SBA are produced annually in Southeast Asian nations [3]. Malaysia alone generates nearly 10 million tons of palm oil residue per year [4]. Therefore, the extraction of SiO_2_ from POFA via methods such as acid leaching [5], alkaline leaching [6], and sol-gel method [7] for utilization as super-hydrophobic coating alleviates the environmental burden brought about by the healthcare and agricultural industry. The super-hydrophobic coating can be successfully utilized for medical implants, contact lenses, controlled drug delivery and diagnostic applications [9,10]. The super-hydrophobicity of the material is primarily utilized to prevent the attachment of microorganisms on the biomedical device’s surface. The super-hydrophobic coatings are therefore able to prevent bacterial contamination and, in turn, eliminate device-associated infections [11,12]. Although the super-hydrophobic coating’s primary function is to prevent bacterial adhesion, the coating should be biocompatible with mammalian cells and must not exhibit cytotoxic properties [13].

In our previous study, we synthesized a super-hydrophobic coating with a WCA of 171° using SiO_2_ extracted from POFA and conjugated it with low surface energy polydimethylsiloxane (PDMS) polymer using hexane as solvent. PDMS has numerous beneficial qualities such as electrical isolating property, high thermal stability, biocompatibility, and low surface energy [22,23]. However, the solvent needs to be substituted with less hazardous materials, as hexane is highly toxic in nature. Therefore, in this study, isopropanol, which is considerably less toxic than hexane was used as a solvent to extract SiO_2_ from POFA. The extracted SiO_2_ was combined with PDMS to produce a super-hydrophobic material and was coated on a glass substrate’s surface. The super-hydrophobic PDMS: SiO_2_ coating was prepared with a ratio of 1:2. The primary objective of this work is to study the effects of PDMS: SiO_2_ surface roughness and surface energy on cell viability, colony formation capabilities and cell proliferation. The cytotoxicity and biocompatibility evaluation were performed via MTT assay, clonogenic assay, and the cell proliferation assay using the mouse fibroblast cell line (L929) and hamster lung fibroblast cells line (V79). The utilization of sustainable POFA waste in this work will reduce the hazardous impacts on the environment and support the Sustainable Development Goals of Malaysia by ensuring sustainable consumption and production patterns.

## 2. Materials and Method

### 2.1. Materials

All chemicals utilized in this study were of analytical grade and utilised as received without further purification. PDMS (average *M*_n_ ~550) with 99% purity was acquired from Sigma Aldrich (Darmstadt, Germany). All other chemicals were obtained from Merck (Darmstadt, Germany). Palm kernels were acquired from Malpom Industries Berhad (Pulau Pinang, Malaysia). Microscope glass slides were purchased from Fisher Scientific (Rochester, NY, USA).

### 2.2. Synthesis of Silica Solution (SS)

The palm kernels acquired locally were thoroughly combusted in a muffle furnace for 6 h with a heating rate of 10 °C/min at 700 °C. The resultant POFA product derived from the combustion was filtered through a 300 µm pore size sieve to eliminate any impurities. Then, 10 g of purified POFA was treated with 100 mL of 1 M NaOH and magnetically stirred for 1 h at 80° before being cooled. The solution was then filtered using a Whatmann filter paper with a pore size of 11 µm and pH adjusted to a value of 3 via drop addition of 1 M H_2_SO_4_. The obtained solution was then designated as silica solution (SS).

### 2.3. Synthesis of PDMS: SS Super-Hydrophobic Solution

The super-hydrophobic solution PDMS: SS was prepared by stirring 4.55 wt% of PDMS (purity of 99% (average *M*_n_ ~550) was purchased from Sigma Aldrich in 100 mL of isopropanol solution for 30 min. Next, SS was slowly added dropwise, and the solution was left to stir for an hour to achieve homogenous dispersion.

### 2.4. Deposition of Super-Hydrophobic Coating on Glass Surface

The synthesized super-hydrophobic solution was uniformly coated on the surface of a pre-treated glass slide using a spray gun with an air pressure of 40 psi from a distance of 20 cm. The freshly coated surface was then dried in an oven at 80° for 5 min. The coating and drying process was repeated 5 more times to obtain an even and thick coating. The coated glass substrate was then completely cured overnight at 80 °C.

### 2.5. Physical Characterisation

The physical properties of the PDSM: SS coated glass slide were analyzed using field emission scanning electron microscopy (FESEM-EDX, Supra 35VP, Zeiss, Oberkochen, Germany), atomic force microscopy (AFM, Nano Navi, SPA400, Seiko Instruments, Chiba, Japan) and goniometer (Model 250-F1, Rame-Hart Instruments Co., Mountain Lakes, NJ, USA). FESEM imaging was conducted with a magnification of 10,000× to obtain clear images of the PDMS: SS super-hydrophobic coating. AFM was conducted in contact mode operation to study the average surface roughness (Ra) and surface energy of the PDMS: SS super-hydrophobic coating in order to understand its effects on biocompatibility. A goniometer was then utilized in this work to ascertain the water contact angle (WCA) and tilting angle (TA) of the synthesized PDMS: SS super-hydrophobic coating and the uncoated glass slide. The samples were carefully placed on a goniometer platform which was attached to the image analyzer. The tests were conducted using sessile drop method which used water droplets with a volume of 5 µL. The WCA and TA were determined using 5 replicates by the Advanced Drop Image software. FESEM was utilized to understand the surface morphology of the PDMS-coated glass slide, which complements the studies on surface roughness, surface energy, water contact angle, and tilting angle.

### 2.6. Cell Line Culturing for Biocompatibility and Cytotoxicity Studies

Mammalian cells were obtained and cultured in this work to study the biocompatibility and cytotoxicity of the PDMS: SS super-hydrophobic coating. Mouse fibroblast cell line (L929 cells, ATCC.CCL1, American Type Culture Collection, Manassas, VA, USA) and hamster lung fibroblast cell line (V79 cells, JCRB0603, Japanese Collection of Research Bioresources Cell Bank, Osaka, Japan) were meticulously cultured in Minimum Essential Medium (MEM) at 37 °C under humidified conditions of 5% CO_2_ and 95% O_2_. The media was further supplemented with 10% of foetal bovine serum (FBS), 10 mL of non-essential amino acid TS (test solution) as well as 100 mmol/L sodium pyruvate TS.

### 2.7. Cell Viability Assay (MTT Assay)

The MTT assay was conducted in technical replicates of 3 by incubating 1 × 10^4^/mL of L929 cells in a 96-well plate at 37 °C with 5% CO_2_ for 24 h. The cell media were then removed from the wells after 24 h incubation and differing PDMS: SS super-hydrophobic coating concentrations of 3.125, 6.25, 12.5, 25, 50, and 100 mg/mL were carefully added into each individual well. Analogous concentrations of positive control, Zinc diethyldithiocarbamate (ZDEC), were also added into the control wells. The test and control extracts were then removed after 24 h of incubation and replaced with 100 µL of growth medium and 100 µL 3-(4,5-dimethylthiazol-2-yl)-2,5-diphenyltetrazolium bromide (MTT). The well plate was then placed in a non-illuminated environment for 4 h at 37 °C. Subsequently, 100 µL of dimethyl sulfoxide (DMSO) and 12.5 µL of glycine buffer were added in sequence into each individual well. The cell viability was then measured by determining the absorbance of light by the cells at 570 nm via a spectrophotometer. The cells were viewed under an inverted phase microscope (Olympus CKX-41, Tokyo, Japan at a magnification of 100×).

### 2.8. Clonogenic Assay

Clonogenic assay was conducted in technical replicates of 3 by incubating and seeding 100 cells/mL of V79 cells in 24-well plates for 24 h at 37 °C under humidified conditions with 5% of CO_2_. Differing PDMS: SS concentrations of 12.5, 25, 50, and 100 mg/mL were then added into the well plates and incubated for six days. The entire procedure was repeated utilizing non-treated cells, 25 mg/mL of ZDEC which served as the positive control and 100 mg/mL of polyethylene (PE) which acted as the negative control. The media were then aerogenated and 1 mL of diluted formaldehyde TS (dFTS) was added to each individual well before incubating it for 30 min at room temperature. Next, the dFTS was disposed and replaced with 1 mL of diluted Giemsa’s TS (dGTS). Then, the well plate was incubated at room temperature for 15 min and the cells were washed with deionized water and air-dried in order to count the colonies. Colonies consisting of more than 50 cells were counted for the clonogenic assay [24].

### 2.9. Cell Proliferation Assay

Cell proliferation assay was conducted in technical replicates of 3 to study mammalian cells that are capable of proliferating but not necessarily capable of colony formation. 1 × 10^4^ cells/mL of L929 cells were seeded in 6-well plates and incubated at 37 °C for 24 h in 5% CO_2_ atmospheric condition. Then, the culture media was substituted with fresh medium containing 12.5 mg/mL of PDMS: SS. Cell proliferation was determined by measuring the total cell number using a haemocytometer at intervals of 1, 3, 5, and 7 days. The culture media were also replaced with fresh media every 48 h in each individual well. The procedure was repeated with non-treated cells, 25 mg/mL of ZDEC (positive control) and 100 mg/mL of PE (negative control). The cells were viewed under an inverted fluorescence microscope (Olympus IX71, Tokyo, Japan, at a magnification of 100×).

### 2.10. Statistical Analysis

All experimental data obtained in this work were analyzed by Student’s *t*-test or Tukey–Kramer, expressed as mean ± standard deviation (SD), and statistical significance was accepted at a *p*-value of less than 0.05.

## 3. Results and Discussion

### 3.1. Effect of PDMS: SS on Water Contact Angle

Bare tile showed a water contact angle of 53° ± 2° (Table 2), implying its hydrophilic behavior. After coating with a PDMS/SS hydrophobic solution, coated samples attain hydrophobic characteristics. It can be seen that the sample coated with PDMS: SS shows the highest water contact angle of 151° and the lowest tilting angle of 7° (Table 2). For the PDMS: SS coated tile, its super-hydrophobicity is attributed to high average surface roughness (Ra) of 21.80 nm, with hierarchical surface morphology as seen in its AFM and FESEM images (Figure 1), indicating hills and valleys, therefore creating air pockets that increase air–liquid contact area [25]. The low tilting angle of 7°, suggests that it is in Cassie–Baxter state as the water droplet can roll off at a low tilting angle. This is probably attributed to its surface morphology with the formation of air voids that reduce the liquid–solid contact between the water droplet and coated surface, leading to rolling effect [21].

### 3.2. MTT Assay

Super-hydrophobic biomaterial coating for applications such as controlled drug release and reduction of bacterial interaction on implants requires the critical evaluation of cell behavior mediated by surface wettability and topography. However, the effects of surface energy and surface roughness on cell viability are often contradictory as cell behavior can be highly dependent on cell type [26]. To comprehensively evaluate the interaction of mammalian cells on the fabricated super-hydrophobic coating, MTT assay was conducted to study the viability of the L929 cell line in differing concentrations of PDMS: SS coating with the utilization of zinc diethyldithiocarbamate (ZDEC) as a positive control. The cells in the MTT assay were treated with PDMS: SS and ZDEC extract with concentrations of 3.125, 6.25, 12.5, 25, 50, and 100 mg/mL for 24 h. The cell viability of L929 after 24 h of stress was analyzed and the results are shown in Figure 2. The MTT assay depends on the reduction of tetrazolium salt to form an insoluble formazan (purple-colored dye) through the aid of dehydrogenase enzymatic activity taking place in the intact mitochondria of an active living or viable cell [27]. This enzymatic activity has a central role in the tricarboxylic acid cycle, as well as in oxidative phosphorylation. Cell viability was quantitatively obtained through MTT assay by dividing the mean optical density (OD) value of the test sample with the mean OD of blank and multiplied by 100.

The assay revealed that the super-hydrophobic coating permits a higher cell viability percentage relative to the positive control material. The MTT quantification assay also suggests that the cell viability may be dependent on the surface energy and surface roughness of the super-hydrophobic coating. The PDMS: SS coating with a concentration of 3.125 mg/mL exhibits a slight cell viability reduction to 92.87% ± 3.33%. This minor change can be attributed to the introduction of the PDMS: SS with relatively high surface roughness and low surface energy. An increment in the concentration of PDMS: SS to 6.25 mg/mL saw a minor improvement to 100.56% ± 7.38% in cell viability in lieu of the increase in quantity. The increase of PDMS: SS coating concentration may have led to a decrease in surface energy due to the higher amount of PDMS. However, surface roughness may also have been reduced due to the higher amount of SS present. This resulted in the availability of a higher surface area for cellular adhesion, leading to improvements in cell viability. This result is in direct agreement with the FESEM images in Figure 3d, which depict a higher number of cells changing to fusiform from spherical, thus validating the increase in contact points between the cells and super-hydrophobic coating. It is attributed to the lowered surface roughness, which dominates among the antagonistic properties.

A similar disposition can be observed in 12.5 mg/mL of PDMS: SS, albeit with slightly more pronounced effects. The concentration yielded a cell viability of 103.51% ± 6.33%. The concentration of 12.5 mg/g has been determined to be the optimal concentration for the highest cell viability. Interestingly, the positive control of ZDEC displays a dramatic decrease to 15.37% ± 0.81% at 12.5 mg/mL, which indicates that ZDEC exhibits significant cytotoxicity at the concentration. As the PDMS: SS concentration is further increased to 25 mg/mL, the cell viability experiences a minor reduction to 96% ± 2.82%. The PDMS: SS concentration of 25 mg/g marks the onset of cell repulsion of the hydrophobic surface. This may be attributed to the agglomeration of the coating on the surface. A drastic decline to 73.98% ± 4.39% was further observed when the concentration of PDMS: SS was increased to 50 mg/mL. At 100 mg/mL of PDMS: SS, it exhibits very high cell toxicity and displays a cell viability of 15.87% ± 0.89%. The cytotoxic surface chemistry and hydrophobicity of the material is primarily responsible for the marked reduction in cell viability. The increase in number of air pockets and air–liquid contact area created by the increased agglomeration may inevitably lead to the reduction in cellular adhesion. Figure 3g,h lend validity to the idea as the FESEM images conclusively show that 50 mg/mL and 100 mg/mL of PDMS: SS actively discourage cellular adhesion on its surface in addition to causing cell apoptosis, as evidenced by the spherical cells. The high cell apoptosis can be attributed to the inability of the cells to attach their extracellular matrix molecules on the surface due to the hydrophobicity of the coating [28,29,30]. On the other hand, the ZDEC coating with concentrations of 25, 50, and 100 mg/mL exhibits cell viability within the low range of 1.37% and 1.72%. The low cell viability of ZDEC at high these high concentrations validates the substantial cytotoxicity to mammalian cells as compared to the PDMS: SS at similar concentrations.

### 3.3. Clonogenic Assay

The independent behavior of cells in regards to biomaterial surface interaction advocates the necessity to ascertain the viability of mammalian cell colonies on PDMS: SS super-hydrophobic coating. This is to ensure that considerable apoptosis of non-hostile mammalian cells does not occur when the coating is in direct contact with human physiology. The capability of cell colonies formation was evaluated through the use of clonogenic assay where V79 cells were treated with PDMS: SS extract at concentrations of 12.5, 25, 50, and 100 mg/mL for six days. In addition to that, clonogenic assay studies were also conducted on non-treated V79 cell line in addition to cell lines treated with PE that serves as the negative control and ZDEC which serves as the positive control. The results are depicted in Figure 3 and Figure 4. The cell colony percentage was determined by dividing the mean number of colonies from the test sample with the number of non-treated colonies and multiplied by 100. The six days of clonogenic assay study revealed the cytotoxic potential of PDMS: SS super-hydrophobic coating with increased sensitivity due to its low densities of cell numbers [31]. The clonogenic assay also highlights the differences between MTT assay and clonogenic assay cell behavior by differentiating the cells which are metabolically active but not necessarily capable of forming [32]. Moreover, the clonogenic assay revealed that the colony formations are highly concentration-dependent. At optimal PDMS: SS concentrations of 12.5 mg/mL, the percentage of viable colonies is 86.9% ± 11.4%.

This is in direct contrast to the positive control which has no viable colonies and therefore has 0 percentage of colony. The relatively high percentage of cell colonies treated with 12.5 mg/mL PDMS: SS can be attributed to the lower surface roughness which dominates the antagonistic low surface energy of PDMS [30]. However, it is of important note that the cell colony percentage in 12.5 mg/mL of PDMS: SS coating is relatively lower than that of non-treated cells. This result is due to the stark difference in surface roughness of the coated and the non-coated surface. It has been well established by previous researchers that smooth and featureless surfaces are more conducive to cell attachment, but this factor also encourages bacterial adhesion on its surface [29]. A mild drop in colony percentage viability to 69.2% ± 9.5% can be observed in PDMS: SS concentration of 25 mg/mL. The drop in viable colony percentage mirrors that of MTT assay as the increase in coating concentration diminishes the contact points and in return this suppresses cell attachment. No viable cell colony can be observed in PDMS: SS coating with concentrations of 50 mg/mL and 100 mg/mL. PDMS: SS concentrations of 50 mg/mL and 100 mg/mL are highly cell repellant surfaces that are incapable of sustaining any cell colonies. Moreover, the cytotoxic surface chemistry of the PDMS: SS coating at those concentrations completely disrupts cellular attachment, and in particular the early extracellular matrix molecules, which leads to premature cell apoptosis [30].

### 3.4. Cell Proliferation Assay

Cell proliferation assay was conducted in this work as a supplementary study to the MTT assay in order to complement the previously obtained results. In the cell proliferation assay, the mammalian cells were treated with PDMS: SS coating possessing a concentration of 12.5 mg/mL for a period of seven days. The cell proliferation assay study was also performed on non-treated cells, as well as cells treated with 25 mg/mL ZDEC (positive control) and 100 mg/mL PE (negative control). The proliferation and growth inhibition of viable cells were meticulously measured at days 1, 3, 5, and 7 after treatment as shown in Figure 5. The FESEM images obtained from the cell proliferation study are presented in Figure 6. The assay results demonstrate that both surface roughness and surface energy did not drastically affect cell adhesion on the first day of study, as evidenced by the lack of major deviations of viable cell count on non-treated cells and studies that utilized PE and PDMS: SS super-hydrophobic coating. However, cells treated with 25 mg/mL of ZDEC reiterated the results obtained in the MTT assay by demonstrating that ZDEC is cytotoxic to mammalian cells at that particular concentration.

In addition to that, the cells will not be able to proliferate at all due to the surface chemistry of the substance even over a period of seven days. On the other hand, the cells treated with PE (negative control) exhibited effective cell growth promotion similar to results seen in the clonogenic assay. The cell viability count showcased remarkable growth kinetics on days 3, 5, and 7 for the negative control as compared to the non-treated cells and cells treated with PDMS: SS coating. This may be attributed to the difference in topography owing to the physical nature of the polymers. Although both PE and PDMS are semi-crystalline in nature, the addition of amorphous SS into PDMS alters its surface topography. Therefore, cells proliferate better on the smooth hydrophobic PE surface as compared to the rough super-hydrophobic PDMS: SS surface [26,33,34,35]. Interestingly, cells treated with PDMS: SS exhibit negligibly higher cell proliferation (229 viable cell count) after seven days as compared to non-treated cells (227 viable cell count). This non-conforming result in comparison to the clonogenic assay test can be ascribed to the nature of cells that behave differently under different conditions [28]. These results suggest that the surface roughness does not affect cell proliferation drastically, as compared to cell colony formation. Moreover, this result reiterates the fact that 12.5 mg/mL PDMS: SS super-hydrophobic coating is conducive for cell proliferation and does not exhibit cell toxicity.

## 4. Conclusions

In summary, a super-hydrophobic PDMS: SS coating was synthesized through the conjugation of PDMS and SS in isopropanol solution. The synthesized coating exhibited remarkable super-hydrophobicity with a water contact angle of 151° which affected mammalian cell behavior in distinct ways. The optimal concentration of 12.5 mg/mL of PDMS: SS yielded the highest cell viability with a value of 103.51% ± 6.33%. The high cell viability on PDMS: SS coating can be ascribed to the relatively low surface roughness which overwhelms the relatively low surface energy of PDMS. On the other hand, 100 mg/mL of PDMS: SS showed marked cytotoxicity with a cell viability of only 15.87% ± 0.89%. Furthermore, optimal PDMS: SS concentrations of 12.5 mg/mL showcased 86.9% ± 11.4% of viable cell colonies due to similar effects of observed in MTT assay at that particular concentration. No viable cell colony can be observed in PDMS: SS coating with concentrations of 50 mg/mL and 100 mg/mL indicating potent cytotoxicity. Moreover, cells treated with 12.5 mg/mL of PDMS: SS exhibited high cell proliferation (229 × 10^4^ viable cell count) after seven days, demonstrating the biocompatibility of the PDMS: SS coating. These factors cement the use of 12.5 mg/mL of biocompatible PDMS: SS super-hydrophobic coating for biomedical applications.

## Figures and Tables

**Figure 1 polymers-12-03034-f001:**
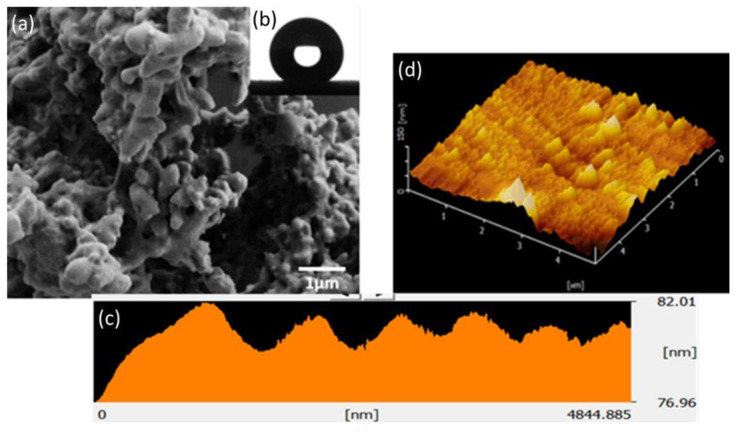
(**a**) FESEM image; (**b**) WCA; (**c**) AFM line profile and (**d**) 3D AFM topographical images of coated samples with PDMS: SS 12.5 mg/L showing hydrophobicity features.

**Figure 2 polymers-12-03034-f002:**
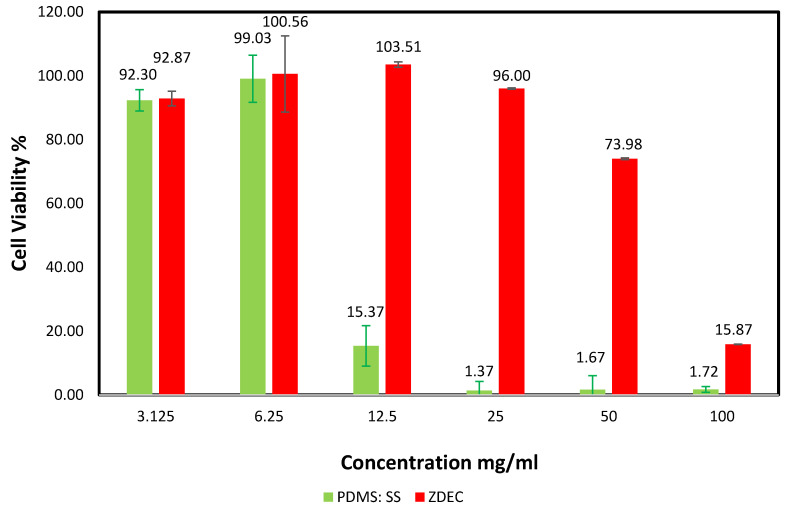
Percentage of L929 cells viability at various concentrations of Zinc diethyldithiocarbamate (ZDEC) and PDMS: SS coating. The experiments were conducted in triplicate independently (n = 3), and the data are expressed as the means ± standard deviation (SD) with *p* 0.05.

**Figure 3 polymers-12-03034-f003:**
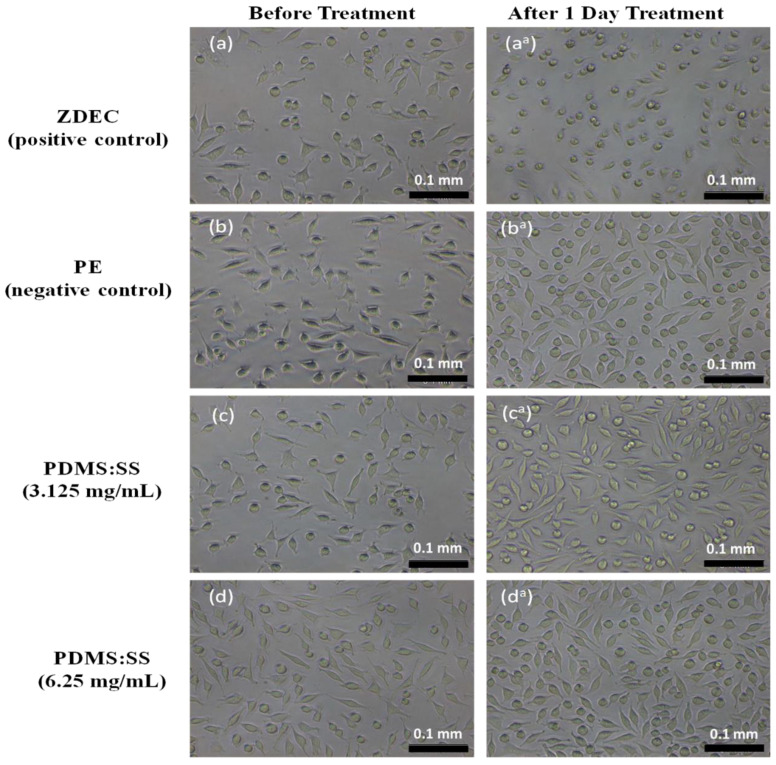
Cell morphology of L929 cells before being treated with (**a**) ZDEC, (**b**) PE, and PDMS: SS coating extract at; (**c**) 3.125 mg/mL; (**d**) 6.25 mg/mL; (**e**) 12.5 mg/mL; (**f**) 25 mg/mL; (**g**) 50 mg/mL; (**h**) 100 mg/mL. Images on the left side show the cell morphology of L929 (**a**–**h**) before treated with the corresponding test items and Images on the right side (**a^a^**, **b^a^**, **c^a^**, **d^a^**, **e^a^**, **f^a^**, **g^a^**, and **h^a^**) show after treated with the corresponding test items (Taken at 100× magnification with a scale bar of 0.1 mm).

**Figure 4 polymers-12-03034-f004:**
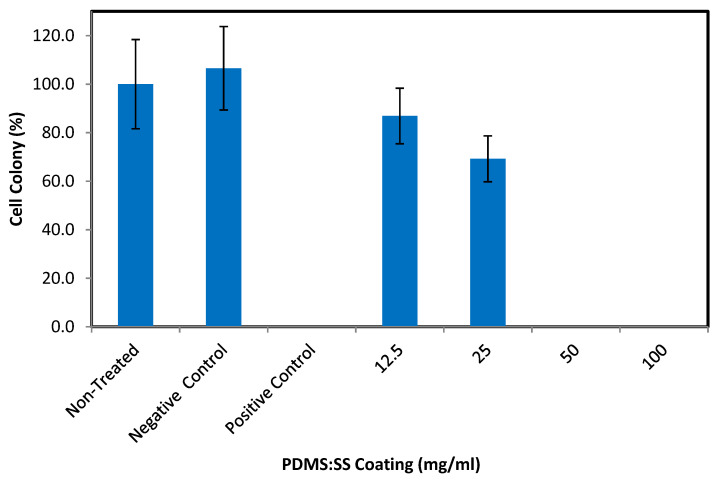
The colony percentage of viable V79 cell. The experiments were conducted in triplicate independently (n = 3), and the data are expressed as the means ± standard deviation (SD) with *p* 0.05.

**Figure 5 polymers-12-03034-f005:**
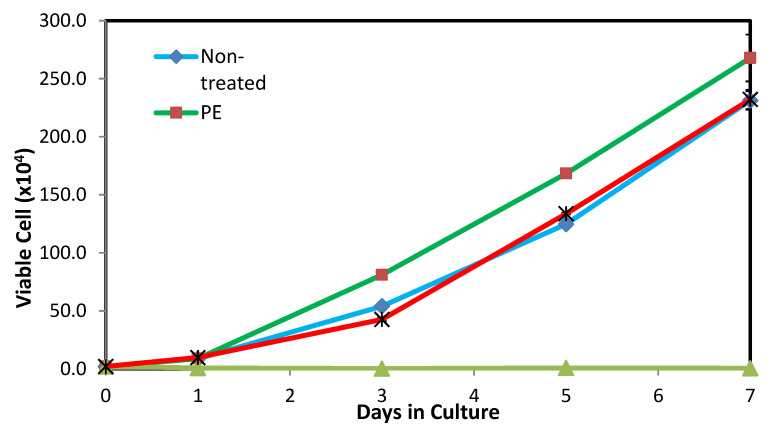
The proliferation curve of viable L929 cell. The experiments were conducted in triplicate independently (n = 3), and the data are expressed as the means ± standard deviation (SD) with *p* 0.05.

**Figure 6 polymers-12-03034-f006:**
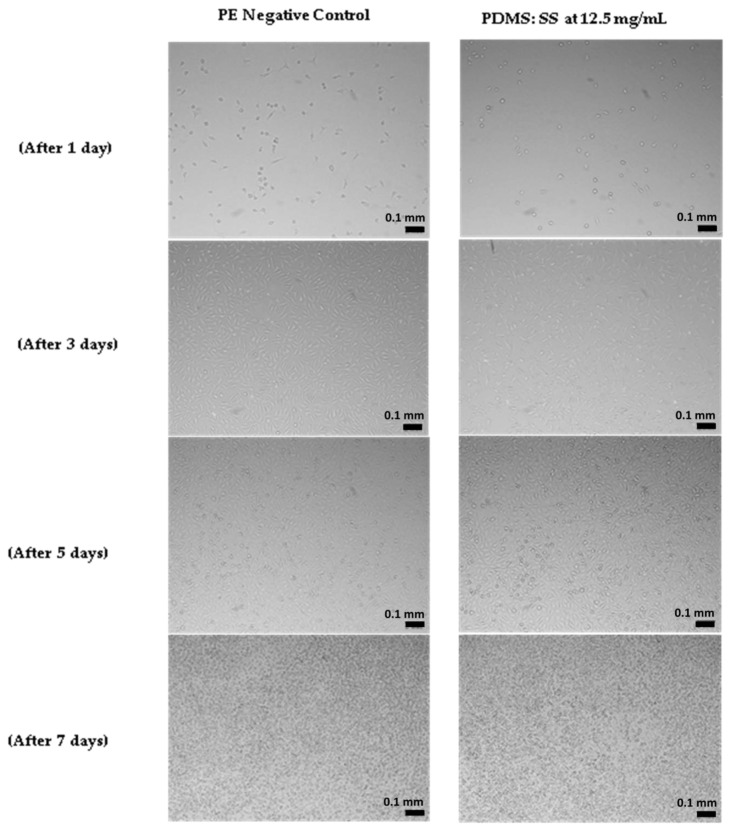
Microscopic images of cells on treatment with PE (negative control) and PDMS: SS coating at 12.5 mg/mL concentration (Taken at 100× magnification with a scale bar of 0.1mm).

**Table 1 polymers-12-03034-t001:** A summary of sustainable waste materials used for super-hydrophobic coating.

Waste Material	Functionalizing Agent	Substrate	Coating Method	WCA (°)	Reference
RHA	1H,1H,2H,2H-perfluorodecyltriethoxysilane	Glass	Spraying	144°	[14]
RHA	1H,1H,2H,2H-perfluorodecyltriethoxysilane	Concrete	Spraying	152°	[15]
RHA	Hydroxyl silicone oil	Glass	Spraying	160°	[16]
RHA	Vinyl triethoxysilane	Porous silica	-	158°	[17]
FA	Dodecyltrimethoxysilane	Cotton textile	Dip coating	152°	[18]
SBA	Dimethyldiethoxysilane	Tiles	Drop-casting	135°	[19]
PSA	Stearic acid	Disc	Pressing	153°	[20]
POFA	Polydimethylsiloxane	Glass	Spraying	171°	[21]

**Table 2 polymers-12-03034-t002:** Water contact angles (WCA) and tilting angles (TA) of glass substrates coated with super-hydrophobic solution.

Substrate	WCA	TA
**Bare Tile**	53° ± 2°	-
**PDMS: SS 1:2**	151° ± 1°	7° ± 1°

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
