# Peer review of "Biocompatibility and Cytotoxicity Study of Polydimethylsiloxane (PDMS) and Palm Oil Fuel Ash (POFA) Sustainable Super-Hydrophobic Coating for Biomedical Applications"

_polymers, 2020, doi:10.3390/polym12123034_

Round 1
Reviewer 1 Report
The manuscript itself is interesting and relatively well written. However, there are present some points which significantly decrease the overall quality of the manuscript.
- Used reagents and majority of instruments are of unknown source. According to the guide for authors, there has to be mentioned supplier, city and country of the origin (in case of US also state). In short, part "2. Materials and methods" has to be significantly rewritten.
- Table 2: A value of WCA for sample PDMS:SS 1:2 is not given together with its error.
- Figs 3. and 6.: The bar size is given for each image with very small alphabets to read them. The size of used bars has to be given in figure capptions.
Author Response
We are grateful to you for your insightful comments on this paper. We have revised the manuscript according to your comments. The corrections have been highlighted in red. Moreover, detailed responses for each comments have been provided as well. Please see the attachment.

Reviewer 2 Report
This paper reports an investigation about a biological evaluation of a coating based in PDMS and palm oil fuel ash. Authors evaluated the biocompatibility and cytotoxicity for their potential use in biomedical applications. The paper includes interesting results with suitable experimental design, data analysis and discussion. Therefore, it is recommended for publication in Polymers after major revision indicated below.
INTRODUCTION
- In line 32, one “[ ]” should be omitted when including reference number 1.
- Please include references to support the statements of lines 41-43.
MATERIALS AND METHODS
- Please include the processing parameters for the microscopy evaluation.
- Which parameters for the surface roughness have been analyzed? Please clarify.
- How many replicates did the authors for each measurement?
- Did the authors perform a statistical analysis of the results obtained? If so, it should be mentioned in the Materials and Methods section as a new Section 2.9. However, if not, this statistical analysis should be performed.
RESULTS AND DISCUSSION
- Is the surface roughness included in Section 3.1 referred to Ra, Rz, Sa or Sz? Please clarify since the meaning of each parameter is different.
- Include error for the WCA and TA values of the system PDMS:SS shown in Table 2.
- Improve the quality of Figure 2, 4 and 5.
REFERENCES
- Authors should include more recent references since there are only 4 from the last three years (2018-2020).

Author Response

(The authors gave the same response as above.)

Round 2
Reviewer 1 Report
The manuscript after its revision is ready to be accepted for publication.
Reviewer 2 Report
Authors performed all the suggestions of the previous review process. Therefore, I recommend this manuscript for publication.